# Predicting Individual Well-Being in Teamwork Contexts Based on Speech Features

**Tobias Zeulner** [1], **Gerhard Johann Hagerer** [1], **Moritz Müller** [1], **Ignacio Vazquez** [2] **and Peter A. Gloor** [3,*]

1   Research Group Social Computing, Technical University of Munich, 85748 Munich, Germany; tobias.zeulner@tum.de (T.Z.); jhagerer@mytum.de (G.J.H.); moritz96@mit.edu (M.M.)
2   System Design and Management, Massachusetts Institute of Technology, Cambridge, MA 02142, USA; ignaciov@mit.edu
3   Center for Collective Intelligence, Massachusetts Institute of Technology, Cambridge, MA 02142, USA
*   Correspondence: pgloor@mit.edu

**Abstract:** Current methods for assessing individual well-being in team collaboration at the workplace often rely on manually collected surveys. This limits continuous real-world data collection and proactive measures to improve team member workplace satisfaction. We propose a method to automatically derive social signals related to individual well-being in team collaboration from raw audio and video data collected in teamwork contexts. The goal was to develop computational methods and measurements to facilitate the mirroring of individuals' well-being to themselves. We focus on how speech behavior is perceived by team members to improve their well-being. Our main contribution is the assembly of an integrated toolchain to perform multi-modal extraction of robust speech features in noisy field settings and to explore which features are predictors of self-reported satisfaction scores. We applied the toolchain to a case study, where we collected videos of 20 teams with 56 participants collaborating over a four-day period in a team project in an educational environment. Our audiovisual speaker diarization extracted individual speech features from a noisy environment. As the dependent variable, team members filled out a daily PERMA (positive emotion, engagement, relationships, meaning, and accomplishment) survey. These well-being scores were predicted using speech features extracted from the videos using machine learning. The results suggest that the proposed toolchain was able to automatically predict individual well-being in teams, leading to better teamwork and happier team members.

**Keywords:** explainable AI; multi-modal speaker diarization; affective computing; social signal processing; team collaboration





## 1. Introduction

Employee well-being has gained increasing importance in recent years, as evidenced by the World Health Organization (WHO) classification of burnout as a medical condition resulting from chronic workplace stress in the 11th Revision of the International Classification of Diseases [1]. This change in classification by a globally recognized health authority acknowledges the impact of chronic stress in the workplace and promotes awareness and research into preventative measures and supportive policies.

The majority of studies assessing and focused on improving well-being rely on surveys, which impose a burden on employees due to the time required for completion. This limitation hinders frequent assessment of well-being, which is crucial for improvement. A potential solution to this limitation is to assess well-being in real time and provide individuals with feedback, a process referred to as virtual mirroring [2].

To realize this vision, it is first necessary to develop a method to identify relevant communication features, e.g., speech patterns, relating to individual well-being during collaboration [3]. These features can then be reflected back to employees, enabling them to become more aware of their well-being. This process can lead to behavioral changes that

enhance happiness and the quality of relationships with others [2]. The primary objective of this study was to determine the appropriate computational methods and measurements to facilitate the implementation of virtual mirroring in the context of employee well-being.

Our contribution is the creation of a noise-robust multi-modal speaker diarization tool applied to a curated dataset of 20 teams collaborating on a creative task for four days in an educational environment. Our tool accounts for background noise and determines speaker identities through speaker diarization for teams working in close proximity. This tool facilitates the multi-modal analysis of speech signals captured in noisy field settings. Privacy concerns are addressed by identifying and removing the data from individuals who want to opt out of data collection. To encourage further research, the algorithms are implemented in a modular design, and the output is standardized using a rich transcription time marked (RTTM) file. Further, an automatic feature extraction pipeline was developed. This pipeline computes a plethora of speech features per individual, time series as well as aggregates, throughout each teamwork session. Lastly, predictive models were developed to predict well-being based on speech features extracted from teamwork sessions, using the positive emotion, engagement, relationships, meaning, and accomplishment (PERMA) framework developed by Seligman [4] as a theoretical foundation. By leveraging SHAP explainability, this contribution advances the understanding of the relationship between speech features and individual well-being, informing potential interventions and strategies to enhance well-being in teamwork settings.

All of our code, including the speaker diarization, automatic feature extraction pipeline, and best models, is publicly available (https://github.com/Zeulni/wellbeing-audio-analysis, accessed on 5 March 2024).

In this study, the authors answer the following research questions:

**RQ1.** What are the challenges of individual well-being prediction in team collaboration based on multi-modal speech data? How can they be addressed?

**RQ2.** Based on our own data, what algorithms and target labels are suitable for predicting well-being in teamwork contexts based on multi-modal speech data?

**RQ3.** Based on our own data, which speech features serve as predictors of individual well-being in team collaboration?

## 2. Related Work

We have drawn on related work in three areas: data collection, data preparation using speaker diarization, and data analysis to predict and understand individual well-being.

### 2.1. Onsite Team Collaboration Data

Numerous studies have examined data collection techniques for team collaboration, with most focusing on recording at least two people using different sensing modalities [5–9]. For example, Ringeval et al. [5] used a webcam setup to record video and audio for dyadic discussions. Oxelmark et al. [6] surveyed participants via interviews to analyze teamwork among students but did not include video or audio data. Koutsombogera and Vogel [7] used surveys and multi-modal sensors, including video and audio, to analyze teamwork, but the camera setup was complex and not easily replicable in a large-scale setting. Similarly, Sanchez-Cortes et al. [8], Braley and Murray [9] analyzed teamwork using a similar multi-modal approach, but in a controlled environment where participants did not have real-world projects to work on, but rather the used a winter survival task commonly used for teamwork experiments, which tests leadership skills in challenging circumstances. All of these experiments were conducted in a controlled environment, and a new setup was needed to collect data in an in-the-wild environment where participant behavior is less predictable. While some experiments have been conducted in the wild, such as in Christensen and Abildgaard [10], in which one team was observed at their work site for several weeks using expensive cameras, these experiments are not scalable for many teams. Ivarsson and Åberg [11] also used audio and video sensors to analyze teamwork in the wild, but in a different setting, i.e., hospital operating rooms. Additionally, video analysis

in education has a long history, for improving teacher education and student learning effectiveness, although most of this video analysis has been performed using manual analysis [12]. More recently, automated video analysis technologies have been used to increase educational effectiveness in the classroom [13].

### 2.2. Multi-Modal Speaker Diarization

The application scenario defines which signal can be used to determine "who spoke when". In our case of team collaboration, all team members are sitting at one table. Thus, the most reliable and least intrusive way of data collection uses one 360° room camera and a room microphone. Classical speaker diarization systems rely solely on audio data to accomplish this task, such as the state-of-the-art approaches for meetings presented by Kang et al. [14] and Zheng et al. [15]. However, these approaches leverage multi-channel audio files, so they are not suitable for data recorded with a single microphone. Moreover, neural-network-based speech separation methods such as deep clustering [16] and permutation invariant training [17], which are part of the state-of-the-art speaker diarization pipeline, cannot properly account for background noise [18]. These and other single-channel speech separation techniques, such as the one presented by Luo and Mesgarani [19], do not effectively handle speech overlaps in realistic multi-speaker scenarios [20], making them unsuitable for field settings. In contrast, utilizing a multi-modal approach by adding video data has been shown to outperform audio-only analysis in the field of speech processing [21]. Thus, recent studies have often leveraged a multi-stage approach and incorporated video data, rather than using a single end-to-end model [22–27]. The method proposed by Yoshioka et al. [22] uses face tracking and identification, sound source localization, and speaker identification, yet it requires multi-channel audio input. Another method, initially introduced by Nagrani et al. [23], first performs face detection and tracking, and then uses active speaker detection (ASD) to determine the synchronization between the mouth movement and speech in the video to identify the speaker. After that, the approach uses face verification to classify whether the recognized face belongs to a specific individual. However, this requires prior knowledge such as the number and images of the speakers, see also Chung et al. [24,27]. To solve this problem, this step was replaced by the clustering of face tracks in the latest version [25]. This identifies all visible speakers in a video based on face embeddings and does not rely on any prior information. However, the integrated ASD step requires two models and does not use all available information, such as the temporal context. Xu et al. [26] used audiovisual information for the final clustering without comparing it with state-of-the-art face recognition systems, which could decrease the model complexity. However, we have not yet seen a publicly available speaker diarization system suitable for noisy field environments with low overall complexity.

### 2.3. Individual Well-Being Data Analysis

The most common method to measure well-being is the use of a survey [28]. Prior research analyzed various predictors of subjective well-being utilizing this method, e.g., personality [29], emotions [30], neuroticism [31,32], or health [33,34]. None of these studies focused on teamwork contexts. Measuring individual well-being in the workplace is a topic that has been widely studied [35,36]. These studies have relied on surveys, which can be time-consuming and introduce bias. Various sensors can be leveraged to collect data and automatically predict well-being, e.g., motion and temperature sensors in smart-home environments [37,38] or smartphone application usage [39–41]. Another common method for predicting well-being is to only use video as a modality, as was done by [42]. However, these approaches do not include speech features.

To predict well-being in the context of depression, the researchers [43–45] used speech features obtained from audio and video, such as, interviews and reading tasks. Huang et al. [46], Kim et al. [47] focused specifically on speech features. Regarding work-related environments, Kuutila et al. [48] used software repositories to predict well-being without

collecting audio data. Izumi et al. [49] took a multi-modal approach including audio and speech data. In summary, there has been research on automated well-being prediction in work contexts leveraging speech features. However, we identified a lack of research and related tooling focusing on speech features.

## 3. Study Design

This section presents the data collection; extraction of speech features, e.g., emotions or communication patterns; and data analysis, relating speech features to PERMA well-being scores.

### 3.1. Teamwork Setting

#### 3.1.1. Study Context

Anonymous Institute (AIA) offers the Anonymous Study (AS) program [50], combining engineering with management sciences, educating early and mid-career professionals to become technically skilled leaders in their organizations [50]. As part of this course, students participate in a yearly workshop. This event spans five days and includes a workshop where selected companies present projects on the first day. Students form teams to work on these projects for the remaining four days, with daily milestones determining team performance.

#### 3.1.2. Teams

Team sizes varied from two to five members. Out of 82 students, 56 signed the informed consent form, thus agreeing to join the video analysis experiment. The teams were formed based on their project preferences and thus were allowed to work on their preferred projects rather than being assigned to a completely unfamiliar project. Participation in our analysis was voluntary and had no influence on the participants' academic grades.

To tackle their projects, the teams had access to a large monitor, a whiteboard, and their personal laptops. They mostly worked together at the table, occasionally standing up to use the whiteboard. Although each team worked on a different project, the tasks were the same throughout the week (e.g., stakeholder analysis). Only teamwork sessions, which lasted between 10 and 90 min each, were considered for our analysis of well-being. These sessions accounted for about 25% of the daily schedule, the remaining time was allocated to workshops. The daily program consisted of a mix of workshops and teamwork, with the duration of each component varying from day to day.

The teams were physically co-located in a single room but visually separated by whiteboards, as shown in Figure 1.

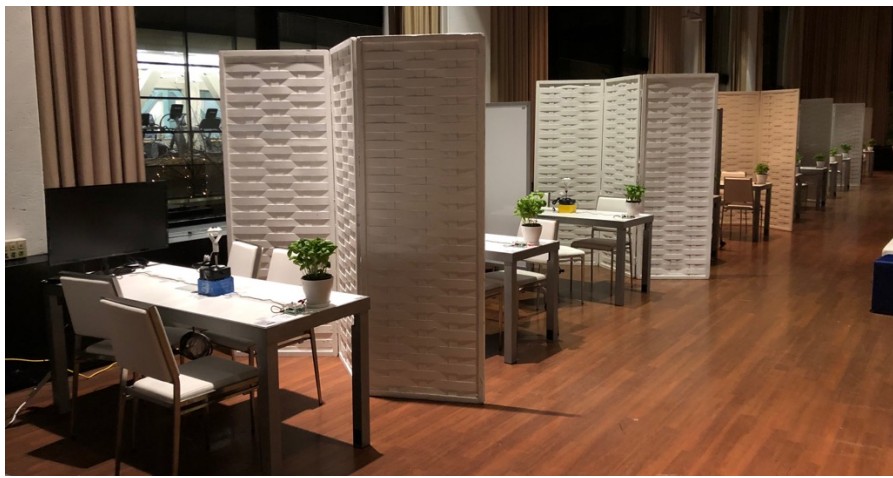

**Figure 1.** Image showing the co-located tables of some teams.

### 3.2. Data Collection

3.2.1. Recording

For this study, we collected multi-modal video and audio speech data. Video was necessary to support speaker identification in noisy environments. In order to minimize intrusiveness for the participants, the video and audio recordings were not conducted individually for each team member, but with one camera and one microphone per team. To capture the faces as best as possible, one 360° camera was used for each team, placed in the center of each table, see Figure 1. We used a 360° camera JVCU360 from j5create to record at a frame rate of 30 fps in HD [51], resulting in one video per team. To achieve a balance between audio quality, intrusiveness, and affordability, one omni-directional conference microphone from TKGOU was placed in the center of each team's table, see Figure 1. The audio quality was degraded by the high level of background noise, which presented challenges for the subsequent data cleansing. To record the captured audio and video, we opted for Zoom, a video conferencing software that is widely used in research [52]. Zoom's intuitive interface and two important features, noise reduction and access to unlimited cloud storage via the Enterprise license, were particularly advantageous for our experiment. Recordings were streamed directly to the cloud, stored in MP4 format, and could be accessed later, since the mapping between the meeting link and the corresponding team was stored locally. Zoom itself was run on Intel NUC mini-computers placed on the tables.

3.2.2. Surveys

To answer the research questions, it was necessary to collect data on the subjective well-being of the participants using the PERMA framework. We relied on the PERMA questionnaire from Donaldson et al. [53] (29 questions), since it is validated, short, and minimizes intrusiveness for participants. It has also frequently been used in workplace environments. It is based on the PERMA+4 model, accounting for additional variance in work-related well-being and performance through the use of four additional components: physical health, mindset, environment, and economic security. To further minimize intrusiveness, we only included the five common PERMA pillars (P Positivity, E Engagement, R Relationships, M Meaning, A Accomplishment) in our questionnaire consisting of 16 questions. Each question was rated on a Likert scale from 1 to 7. It took between 3 and 5 min to complete the online survey.

In total, data were collected and analyzed from 56 students over four days, and the PERMA survey was answered by an average of 52 students per day. All 56 participating students signed the consent form. Out of 56 participants, 20 completed the personal questionnaire. Thirteen of the participants identified as male, and seven identified as female. Ten participants were between 31 and 35 years old, five between 26 and 30, four between 36 and 40, and one between 46 and 50. The ethnicities of the participants who completed the questionnaire were "White/Caucasian" with eight participants, "Asian/Pacific Islander" with seven, "Hispanic" with four, and one "Multiple ethnicities". Eleven people indicated their professional role as "Engineer", two as "Analyst", "Project Manager", and "Other", and one as "Scientist". At the team level, data were collected and analyzed from 20 teams.

### 3.3. Data Preparation

The main focus of this study was on analyzing the speech data collected during the workshop week. The preparation and extraction of the following speech features will be described subsequently.According to Wilson [54], happiness is related to being extrovert and vocative, and extroverts generally talk more than introverts [55]. Thus, speaking duration can serve as an indicator of well-being and was therefore included as a feature in this study. This included the speaking time in seconds, as well as the number of utterances. As extroverts tend to be more likely to interrupt others in conversations, whereas introverts tend to avoid such behavior [56], we used the number of interruptions as an additional audio feature. Last but not least, since positive emotion is considered the first pillar of

the PERMA model, this study focused on the calculation of emotional features as the first component of the analysis.

### 3.3.1. Data Preprocessing

The first step of data analysis involves data reduction, which is done specifically for video and audio data types. Based on our notes taken during the Workshop Week, the start and end times in seconds of the sessions were captured for each team in a spreadsheet, which enabled the automatic extraction of the sessions using a Python script and the MoviePy library (https://zulko.github.io/moviepy/). No data were extracted on Wednesday of the Workshop Week, as no teamwork took place that day. After reducing the MP4 data (recorded with Zoom) for 20 teams, a total of 93:06 h of data remained, representing 4:39 h of collaboration per team and approximately 1:33 h of collaboration per team per day. Across all teams, the minimum duration of the collaboration over the three days was 3:51 h and the maximum was 4:59 h. The standard deviation equaled 15.4 min.

### 3.3.2. Speaker Diarization

The speaker diarization model takes an MP4 file as input and outputs an RTTM file for a given video. The implementation of each algorithm in the pipeline is explained below.

#### Face Detection

The model weights and code for the face detection model single shot scale-invariant face detector (S3FD) were taken from a publicly available GitHub repository (https://github.com/cs-giung/face-detection-pytorch. It was trained on the WIDER FACE dataset [57]). To reduce the runtime by a factor of 2, we updated the code to track every second instead of every frame, which was implemented throughout the pipeline. Although no quantitative measures were taken to evaluate the exact impact of this change, a qualitative evaluation was performed, which is described in Section 3.3.4.

#### Face Tracking

The face tracking algorithm used in our study is based on the code from Chung and Zisserman [58], which is given in a publicly available GitHub repository (https://github.com/joonson/syncnet_python). We determined the parameters of this rule-based face tracking algorithm based on the qualitative audiovisual speaker diarization evaluation in Section 3.3.4 as follows: To assign a bounding box to a track, we set the threshold for intersection over union (IOU) to 0.3. We reduced this threshold in comparison to the one used for ASD (https://github.com/TaoRuijie/TalkNet-ASD), which is 0.5, to maintain high tracking accuracy, while reducing the number of tracks. The threshold for terminating a track after a face was no longer detected was set to 100 frames (i.e., 4 s in a 25 frames per second (fps) video). The minimum number of faces per track was set to 25, as opposed to the 10 used for ASD (https://github.com/TaoRuijie/TalkNet-ASD), to exclude short video tracks. Consistently with the original code, the minimum size of each detected face was one pixel.

#### Face Cropping

The original code (https://github.com/TaoRuijie/TalkNet-ASD) implements a loop for each track of the input video to crop the faces, which results in a significant increase in runtime with the number of tracks. To mitigate this problem and optimize the algorithm, a new approach was developed, looping over the input video only once and directly cropping the faces for all tracks.

Before saving the files, several image transformations were performed, including resizing and grayscale conversion. The order of these transformations was based on the original code (https://github.com/TaoRuijie/TalkNet-ASD).

Active Speaker Detection

For the active speaker detection, the code from Tao et al. [59] (https://github.com/TaoRuijie/TalkNet-ASD) was used as the base. However, as mentioned earlier, a downsampled version of the video data was used in our study, with every second frame being processed. To ensure that the output vector of the ASD model represented the original timeline, the lengths of both the video and audio data were doubled before feeding them into TalkNet. This was achieved by replacing the skipped frames in both the video and audio data with the data from each preceding frame.

Scores-to-Speech Segment Transformation

For the average smoothing algorithm, a window size *k* of 5 was used, which had been also chosen in the original code (https://github.com/TaoRuijie/TalkNet-ASD) to visualize the results.

Face-Track Clustering

For each track *i* containing the face of one person over time, ten face embeddings for randomly selected images *j* were stored, denoted by $\mathbf{e}^{i,j}$. To improve the robustness, an average face embedding with a size of 512, denoted by $\mathbf{e}_i$, was calculated for track *i* as follows:

$$\mathbf{e}_i = \frac{1}{n_i} \sum_{j=1}^{10} \mathbf{e}^{i,j} \cdot [s^{i,j} > 0.65], \tag{1}$$

where $n_i$ is the number of embeddings with a detection probability greater than 0.65 for track *i*, and $[s^{i,j} > 0.65]$ is an indicator function that evaluates as 1 if the detection probability $s^{i,j} > 0.65$, and 0 otherwise. If the indicator function evaluates as 0 for all ten images, the track is discarded, as a high-quality face embedding cannot be guaranteed. The detection probability is provided by the face embedding model and is a score between 0 and 1, where 1 means the highest confidence in the face detection decision. With the threshold set to 0.65, only high-confidence face embeddings are used to calculate the average embedding, resulting in a more robust average embedding. The threshold is manually set to 0.65 based on the qualitative audiovisual speaker diarization evaluation, which is described in Section 3.3.4.

To cluster the face tracks with density-based spatial clustering of applications with noise (DBSCAN), cosine similarity is used as a distance measure. The choice of a threshold is critical for determining clusters of embeddings. After running several tests, a threshold of 0.3 was found to be optimal.

RTTM File

Each person identified in the input video is assigned a unique ID in the RTTM file, and an image containing the ID as a filename is stored in a folder for identification purposes. At the end of the pipeline, a single file is created for each input video, allowing the calculation of individual audio features. Since it uses a standardized format, this part is modular and can be easily replaced by other speaker diarization methods that output an RTTM file, while the subsequent parts can remain unchanged. This file standard is commonly used for speaker diarization tasks and was described by Ryant et al. [60]. Nevertheless, the generated file lacks information about the presence or absence of people in the meeting. In cases where a person does not speak in part of the meeting because they are not present, they are still counted as if they are present but not speaking. Thus, once a person is mentioned in the output file, he or she is assumed to be present for the entire duration of the meeting.

We used standard Python to calculate the speaking duration, number of utterances, and number of interruptions. These three values were calculated in an absolute and relative manner, resulting in six speech features overall.

### 3.3.3. Audio Feature Calculation

We used standard Python to calculate the speaking duration, number of utterances, and number of interruptions for each session, respectively. These three values were summed per session in an absolute manner. Additionally, a relative number was derived for each feature by comparing the value of a team member to the respective sum of all team members. This resulted in six speech features overall. We approximated interruptions as defined by Fu et al. [61], by checking two conditions: (a) the interrupter starts speaking before the interrupted has finished and (b) the interrupter stops speaking after the interrupted. In addition, three emotional features, i.e., valence, arousal, and dominance, were derived from our speech audio signals using the wav2vec 2.0 model provided by Wagner et al. [62] on their GitHub repository (https://github.com/audeering/w2v2-how-to) Using such a deep learning model for feature extraction from raw speech data has been shown to be beneficial, as it is not constrained by existing knowledge of speech and emotion [63]. The utilized approach allows the model to base its predictions solely on the content of what is said, rather than how it is said. The output layer is a three-dimensional vector, where each dimension corresponds to one of the three dimensions of emotions (arousal, dominance, and valence) and has a value between 0 and 1. We chose a time window of five minutes for our study, for each of which nine time series were computed, i.e., one for each feature. Each time series was aggregated as explained later on.

### 3.3.4. Qualitative Assessment of the Software

In order to find the right parameters for the audiovisual speaker diarization and the subsequent time series feature calculation, a manual qualitative evaluation was conducted, since a complete annotation of the whole dataset for tuning each parameter was not practically feasible. Therefore, a random 30 min video representative of the dataset was selected. The video included a team of four individuals of different ethnic backgrounds and genders, including one female and three males. It should be noted that the disadvantage of using a video from the database for the evaluation was that the content could not be controlled, while the advantage was that it represented the original environment, including factors such as background noise, video quality, and lighting conditions.

Audiovisual Speaker Diarization Evaluation

The accuracy of the audiovisual speaker diarization was evaluated by examining the output of the RTTM file. The evaluation criteria chosen were whether the speech segments were assigned to the correct person and whether they matched the actual speaking segments of that person. To this end, software was written to graphically highlight the speech segments for each team member using a bounding box calculated by face recognition. The software colors the bounding box red when the corresponding person is not speaking and green when they are speaking. The track ID of each speech segment is also displayed so that which cluster or person it is associated can be analyzed.

The first step in the evaluation process involved the assessment of two criteria for the speaker diarization algorithm *without* skipping frames, i.e., the vanilla version. Qualitative analysis indicated that the accuracy in detecting speech segments of a person was comparable to that reported in the original study of the ASD model by Tao et al. [59]. In most cases, the software was able to detect when a person was speaking, with speaking segments interrupted by short pauses in sentences. However, the accuracy of the algorithm decreased in poor lighting conditions or when only part of a person's face was visible. In addition, if a person's face was not visible because they were writing on a whiteboard or turning their back, no data could be computed because the ASD model relies on the person's frontal face. The accuracy of the face verification and clustering algorithms used to match speech segments to specific individuals was high, even for individuals wearing face masks. However, the accuracy decreased in low light conditions or when only part of a person's face was visible. In addition, it also declined as more tracks are created, because of the increased chance that some outlier tracks could be merged with the wrong

cluster (i.e., the wrong person). Many tracks occurred when a person's face reappeared frequently during the session, e.g., when they frequently left the camera's field of view and then returned.

Next, the evaluation criteria were examined, *with* frames skipped to shorten the runtime. The accuracy of the algorithm decreased when tracking every third frame, especially for short speech segments. This was likely due to the dynamic nature of the speech, which required a finer granularity in time prediction [59]. However, when skipping every second frame, no noticeable drop in accuracy was observed in the visual evaluation. This was especially true for the face verification algorithm, which does not depend on frame-level granularity. A decrease in accuracy was observed for very short speaking segments below 0.1 s, which were not detected by the software. This observation was made by comparing the two output RTTM files. However, since such segments were rare and the observed difference was small, this method was chosen for its shorter runtime. Hence, the runtime was approximately halved, which was a crucial step towards the research team's goal of achieving real-time performance.

Audio Feature Calculation Evaluation

The evaluation criteria for the accuracy of audio analysis were based on a binary classification of whether the computed features matched the ground truth or not. To assess the accuracy of the computed features, a sample RTTM file was created, which was used to test the algorithms. Since the ground truth of the speaking duration, the number of utterances, and the number of interruptions for this file were known, the accuracy of the algorithms could be determined.

Finally, the emotional content of the video was assessed using a qualitative evaluation approach. The criteria for the evaluation were based on the identification of scenes in which the model was expected to produce high or low arousal, valence, and dominance values based on linguistic information. The output of the voice emotion recognition (VER) model was then compared to the expected values and evaluated accordingly. The evaluation was performed for the entire 30 min video. Our qualitative judgment indicated that the model performed best for arousal, consistent with the results by Wagner et al. [62]. However, not all identified scenes showed a significant increase or decrease in emotion. This may have been due to the limitations of the performance measured for each of the three emotion categories described in Section 3.3.3 and the fact that the current pipeline does not filter out overlaps. Consequently, the audio clip of a speech segment could also contain the voice of another person speaking at the same time.

3.3.5. Feature Extraction, Data Cleaning, and Feature Engineering

The following section first describes the full pipeline, consisting of feature extraction, data cleaning, and feature engineering. This is presented in Figure 2. To curate the dataset that served as input to this pipeline, we extracted the nine time-series features from the MP4 files of the 17 teams that were included in the final dataset. The students who did not sign an informed consent form were excluded from the feature extraction and analysis using our visual speaker identification pipeline. Subsequently, all teamwork sessions in a day were concatenated and assigned to the corresponding daily PERMA scores, as scores were collected at the individual level and on a daily basis. In this way, this study assumed that only the teamwork sessions (and not the other workshop sessions) were reflected in the well-being scores. In addition, each day was considered independently of the others for each speaker, resulting in up to three data points per speaker, as the teamwork sessions occurred on three out of four workshop days. This left a total of 87 data points.

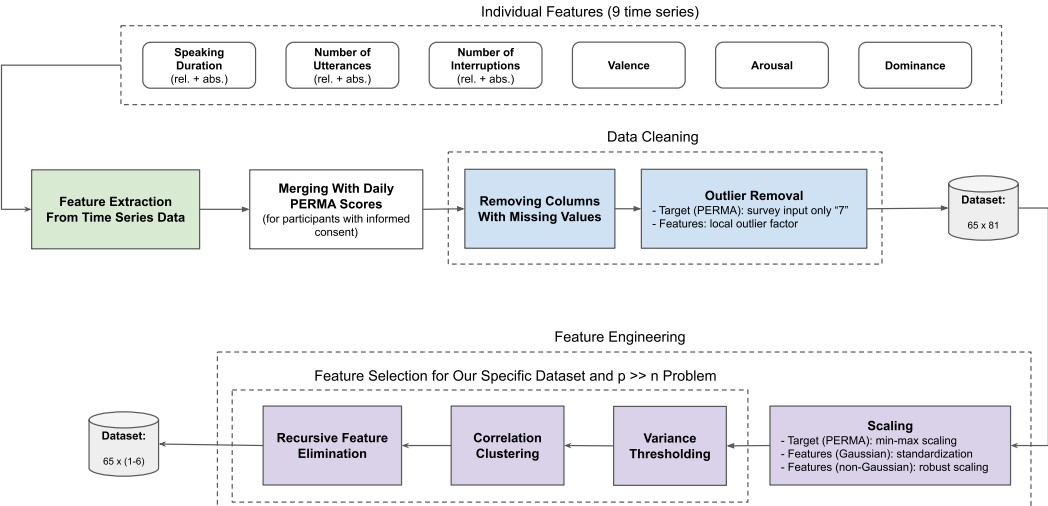

**Figure 2.** An overview of the entire pipeline consisting of feature extraction, data cleaning, and feature engineering.

Feature Extraction from Time Series Data

To use our set of time series $D = \{\chi_i\}_{i=1}^{9}$ as input to the supervised machine learning algorithms, each time series $\chi_i$ had to be transformed into a feature space with a defined dimensionality $m$ and a feature vector $x_i = (x_{i,1}, x_{i,2}, \ldots, x_{i,m})$ specific to the problem at hand [64]. In this study, two approaches were used, differing in the size of $m$. In the first approach, the desired features were manually selected, resulting in the implementation of nine features ($m_{small} = 9$), namely mean, median, 25% percentile, 75% percentile, minimum value, maximum value, standard deviation, variance, and the slope of each time series. The latter was calculated by fitting a linear regression. This approach resulted in a total of 81 characteristics for each individual.

Outlier Removal

Due to the small sample size, the machine learning model was sensitive to outliers. To address this issue, we removed outliers from both the target variable and the feature set. First, we plot individuals' PERMA scores, to identify and remove three participants who completed all 16 questions with the highest possible score. To detect outliers in the feature set, we used the local outlier factor (LOF) algorithm, which is appropriate for high-dimensional data [65]. The algorithm computes a density measure, called the glslof value, by comparing the local density of a sample to that of its neighbors, and identifies potential outliers based on this measure [66]. We used the default number of neighbors for the corresponding scikit-learn function (https://scikit-learn.org/stable/modules/generated/sklearn.neighbors.LocalOutlierFactor.html, accessed on 5 March 2024) (i.e., 20) and set the contamination value to 0.03 assuming a similar amount of outliers as for the target variable. This resulted in the identification and removal of two outliers. Given the training–test split as described above, 65 samples remained after the removal of all identified outliers.

Scaling

Linear regression models are sensitive to the scaling of input features, while tree-based models are not [67]. Since both types of models would be used, feature scaling was required. Scaling is usually performed separately for each feature type [67]. Therefore, different scaling methods were used for the target variable (PERMA scores), features with a Gaussian distribution, and features with a non-Gaussian distribution. For the target variable, two of the PERMA columns had values in the range of 3 to 7, while the range for the remaining three columns was even shorter. Assuming that this distribution was representative of future inference, and with prior knowledge of the lower and upper limits of PERMA scores based on the survey design, we applied min-max scaling from the scikit-

learn library (https://scikit-learn.org/stable/modules/generated/sklearn.preprocessing.MinMaxScaler.html, accessed on 5 March 2024) This scaled the resulting scores to a range between 0 and 1. This involved removing the mean and scaling the data to unit variance. Since the original features followed a Gaussian distribution, the scaled features also followed a Gaussian distribution [67]. Because min-max scaling and standardization are sensitive to outliers [68], robust scaling was applied to features with a non-Gaussian distribution. This involved removing the median and scaling the data according to the interquartile range and was implemented in the scikit-learn library (https://scikit-learn.org/stable/modules/generated/sklearn.preprocessing.RobustScaler.html, accessed on 5 March 2024) [69].

Feature Selection

For each PERMA pillar, a separate classifier was trained based on the automatically selected features shown in Table 1. The first algorithm applied in the feature selection process was variance thresholding. In the second step, correlation clustering was applied to remove redundant features in terms of the Pearson correlation coefficient. The final step in the feature selection process utilized recursive feature elimination (RFE) based on the scikit-learn library (https://scikit-learn.org/stable/modules/generated/sklearn.feature_selection.RFECV.html, accessed on 5 March 2024). Consequently, our approach, with rigorous validation procedures, provided a solid foundation for the robustness of our models. Features selected through RFE might be challenging for the classifier to model and may introduce bias, due to the unique stochastic information in a given feature set. We addressed this problem using leave-one-out cross-validation (LOOCV) (with mean absolute error (MAE) as metric) and L2 regularization during the feature selection process, thus enhancing model robustness. The final number of features was chosen based on the one-to-ten rule and the application of this rule. This suggested a ratio of 10 to 20 between the number of features and samples [70], with the lower limit of features being flexible (since we had 65 samples in the training set, the number of features should be less than seven).

**Table 1.** These are the input features which were used to train the classifier for the respective PERMA pillar. For each pillar, a different set of features was used. The features were selected automatically, as described in Section Feature Extraction from Time Series Data.

| Pillar | # Features | Selected Feature(s) |
|--------|-----------|---------------------|
| P | 2 | norm_num_interruptions_relative_median, arousal_std |
| E | 7 | valence_min, valence_q75, norm_num_interruptions_relative_median, arousal_std, valence_var, norm_num_interruptions_absolute_mean, norm_num_interruptions_relative_max |
| R | 1 | arousal_std |
| M | 6 | valence_min, valence_max, norm_num_interruptions_relative_median, norm_speak_duration _relative_mean, norm_num_interruptions_relative_q25, norm_num_utterances_absolute_max |
| A | 1 | norm_num_interruptions_relative_q25 |

*3.4. Data Analysis*

The computed audio features were correlated with and used to predict well-being, i.e., the PERMA pillars. Further, the feature importance was explored. The three experiments are explained below.

3.4.1. Correlation of Features with PERMA Pillars

The RFE algorithm was used separately for each target variable. Subsequently, the correlation between the selected characteristics and the corresponding target variable was determined for each pillar using the Pearson correlation coefficient. The results were then sorted in descending order, from high positive to high negative correlation.

### 3.4.2. Evaluation of Classification Models

Experimenting with different prediction methods, we found that classification was well suited as a method for predicting the target variables. To this end, the target variables were categorized using percentile-based binning, because the distribution of the target variables was found to be non-uniform. Here, each class represented a percentile range. This means that each pillar was split into two classes, i.e., non-satisfactory vs. satisfactory, with the threshold equal to the median (50% percentile) resulting in two balanced classes. Balanced accuracy was still used as an evaluation measure [71]. In order to train the classification models, the best model per pillar was identified by grid search and LOOCV. The hyperparameters used to train the four selected models are listed in Table 2. For the k-nearest neighbor (k-NN) classifier, the hyperparameters used were the number of neighbors, distance weighting, and distance metric. For the random forest model, class_weight was set to balanced to account for imbalanced datasets, and the number of estimators in the ensemble and the maximum depth were chosen as hyperparameters to prevent overfitting. Similarly, hyperparameters such as learning rate and maximum depth were set for the extreme gradient boosting (XGBoost) model. Since there is no hyperparameter for the maximum depth for the categorical boosting (CatBoost) classifier, the depth was set as a parameter in combination with the learning rate.

**Table 2.** The four classification models trained via grid search with the corresponding hyperparameters.

| Classifier | Hyperparameters |
|---|---|
| k-NN | n_neighbors: [3, 5, 7, 9, 11, 13, 15], weights: ['uniform', 'distance'], metric: ['euclidean', 'manhattan'] |
| Random forests | n_estimators: [100, 200], max_depth: [3, 5, 7] |
| XGBoost | learning_rate: [0.01, 0.1], max_depth: [3, 5, 7] |
| CatBoost | learning_rate: [0.01, 0.1], depth: [3, 5, 7] |

After training the four models for each pillar, the model with the highest balanced accuracy on the validation set was selected as the optimal model per pillar. The final performance of the selected models was evaluated by calculating the balanced accuracy on the test set. The baseline was defined as chance level, i.e., 50% for two classes—see Figure 3. In this context, the performance of the model was expected to outperform this baseline, indicating higher accuracy than random guessing. In that case, it was expected that SHAP values would also indicate relevant features for classification.

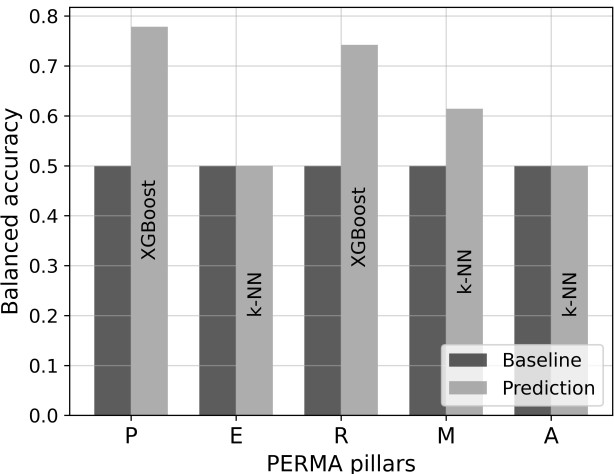

**Figure 3.** Baseline based on chance level vs. prediction balanced accuracy by PERMA pillar on the test subset for the two-class version. For each pillar, the best model from the validation set is shown.

A ratio greater than 1 was expected, indicating better performance than the baseline. The version with the highest average ratio across all pillars was selected as the best performing version.

The scikit-learn library was used to implement the k-NN (https://scikit-learn.org/stable/modules/generated/sklearn.neighbors.KNeighborsClassifier.html, accessed on 5 March 2024) and random forest classifiers (https://scikit-learn.org/stable/modules/generated/sklearn.ensemble.RandomForestClassifier.html, accessed on 5 March 2024). The XGBoost (https://github.com/dmlc/xgboost, accessed on 5 March 2024) and CatBoost (https://github.com/catboost/catboost, accessed on 5 March 2024) models were implemented leveraging the respective Python libraries.

### 3.4.3. Feature Importance of Classification Models

The feature importance values were calculated for each pillar. However, because the feature importance for k-NN could not be calculated directly, Shapley additive explanations (SHAP) values were used to examine the influence of each feature on the model for each class. The SHAP values were calculated for each class, to better interpret which features had which influence on the class assignment. For this, the same library was used as for the regression task.

## 4. Results

The results of the experiments explained previously in Section 3.4 are presented below. The interpretation of these results is given in Section 5.

### 4.1. Selected Features for Classification Models

In Table 1, the number of selected features as well as the feature names are presented for each PERMA pillar in the dataset. These are the input features which were used to train the classifiers for each PERMA pillar. The RFE algorithm identified the most features for pillar E and the fewest for pillars R and A. Some features were selected for multiple pillars, such as norm_num_interruptions_relative_median and arousal_std. Six of the nine original time series features were represented, while the extracted features from the relative number of utterances, absolute speaking duration, and dominance were not selected.

### 4.2. Evaluation of Classification Models

The final performance evaluation of the selected models was performed for each PERMA pillar on the test set after selecting the best models based on their performance on the validation set.

Figure 3 shows a comparison between the baseline and the prediction of the best models with their corresponding balanced accuracies on the test set for the two-class version. The XGBoost is the best performing model for pillars P and R, while the k-NN classifier is the best performing model for the other pillars. For pillars E and A, the prediction accuracy was equal to the baseline. The accuracy of the model for pillar P was 78%, for R 74%, and for M 61%.

### 4.3. SHAP Values of Classification Models

The SHAP values for each PERMA pillar are presented in Figure 4. The interpretation of some features is obvious, such as the 75% quantile of valence for class "satisfactory" and Pillar E (Engagement), where higher values indicated by red colored points signify higher SHAP values and thus to a higher probability of belonging to class "satisfactory". However, the interpretation of other features, such as the variance of valence for the same pillar and class, is not clear.

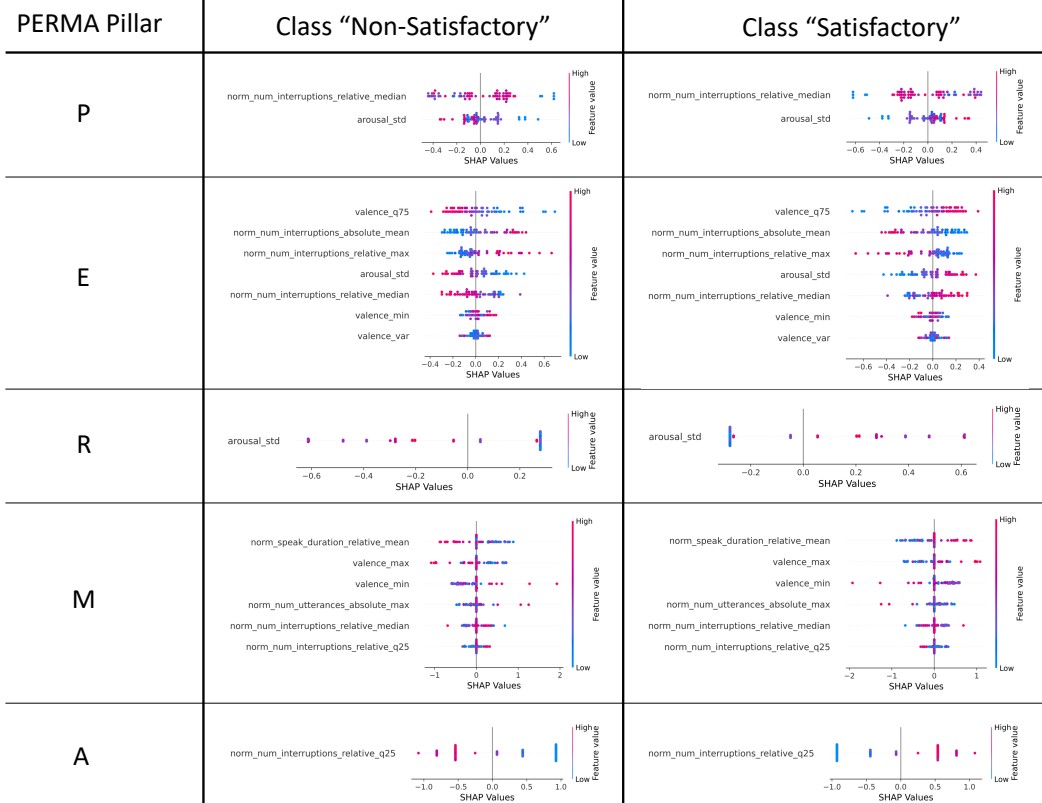

**Figure 4.** The SHAP values of the best performing classification models for the corresponding PERMA pillars.

## 5. Findings

### 5.1. Answers to Research Questions

Three research questions were formulated and addressed in this study. Based on the results presented in the previous chapter, these are answered below.

RQ1. What are the challenges of individual well-being prediction in team collaboration based on multi-modal speech data? How can they be addressed?

We focused on onsite team collaboration in a real-world working scenario, i.e., up to five team members mostly sitting at a table while being engaged in conversations for conceptual work. Some problems encountered and proposed solutions are subsequently listed. To account for multiple individuals whose movements cannot be controlled, it is helpful to use unobtrusive 360° cameras and room microphones for recording. Only signals from individuals who give privacy consent should be considered. This issue was addressed using noise-robust multi-modal speaker identification and diarization to filter out data from individuals who did not give consent. To achieve large-scale well-being data collection without incentives, the reduced and unobtrusive PERMA questionnaire from Donaldson et al. [53] was suitable. To extract affective speech features relevant for well-being, the state-of-the-art wav2vec 2.0 model provided by Wagner et al. [62] extracted dominance, valence, and arousal. To tackle the small number of annotated samples vs. the high number of speech features, feature selection was based on variance thresholding, correlation clustering, and recursive feature elimination. We provide the whole data pipeline as an open source repository along with this publication.

RQ2. Based on our own data, what algorithms and target labels are suitable for predicting well-being in teamwork contexts based on multi-modal speech data?

One goal of this study was to identify the most appropriate method for predicting well-being based on audio data in a team collaboration context. For this purpose, the

performance of classification methods was compared. Classification predicts a category such as "low" or "high" engagement, which is a common practice in sentiment analysis and is shown to be beneficial in the comparison in Appendix A. A single classifier that performed best for all pillars could not be identified. Instead, the XGBoost classifier proves to be optimal for pillars P and R, which also achieved the highest accuracies among all pillars. The maximum depth of the ensembled trees in both XGBoost classifiers was 5, which corresponds to the middle of the specified range. For pillars E, M, and A, the k-NN model performed better. Here, the euclidean metric rather than the Manhattan metric proved optimal for all three pillars, and uniform weights performed better than distance weights.

RQ3. Based on our own data, which speech features serve as predictors of individual well-being in team collaboration?

This study suggested that the standard deviation of arousal was the most important factor for overall well-being, as it was found to be an important feature in three of the five pillars (P, E, and R). In all three pillars, a higher standard deviation led to a higher score on the target variable. However, causality could not be established on the basis of SHAP values alone, and further research is needed to investigate this relationship. Nevertheless, it is possible to hypothesize that high average levels of positive emotions do not necessarily lead to higher well-being. Instead, a high "emotional roller coaster", i.e., strongly varying levels of activity-related affect (arousal), might be related to a more engaged and thus more motivating working atmosphere. This, in turn, might be more likely to promote overall well-being in a professional working environment. This hypothesis is further supported by the fact that two of the three pillars where this feature was present had the highest accuracies and thus reliabilities among all pillars, with 78% and 74% for pillars P and R, respectively.

*5.2. Limitations*

Our study showed limitations both on the technical and the organizational sides.

On the organizational side, positive psychology frameworks, such as PERMA, are known to be sensitive to cultural, gender, and social class differences [72–74]. Therefore, people from different cultures may rate their well-being differently in surveys. This potential source of bias was not considered in this study. In addition, the results cannot be generalized, due to the limited sample size and the specific group of participants.

Another limitation was the microphone placement. While one microphone in the center of the table per team minimized intrusiveness, a single microphone per team member would increase data quality and reduce dependence on the position of the team member in relation to the microphone.

Besides that, both the ASD and face detection algorithms depend on visible and frontal faces, which made it difficult to capture speech segments when the subjects' faces were not visible or frontal. In addition, the performance of the ASD algorithm decreased with more visible faces in the video and fewer pixels per face, as explained by Tao et al. [59]. Another limitation came from creating a single RTTM file per teamwork session, which assumed that a person who spoke once in the session was present for the entire duration. The current system cannot distinguish between a person's absence and their silence during the session.

While a qualitative assessment of the built speaker diarization and audio feature calculation system was conducted, there was a lack of quantitative measures to support the reliability of the system.

When computing emotions such as valence, arousal, and dominance, overlapping speech is not filtered out because single-channel speech separation techniques are not effective in real-world multi-speaker scenarios [20]. Moreover, the study of VER is an ongoing research topic, with challenges such as improving valence performance and addressing generalization and robustness issues [62].

Our definition of interruptions given in Section 3.3.3 could be considered simplistic, whereas Fu et al. [61] also incorporated multi-modal features including video data to decide "who attained the floor".

Moreover, features such as interruptions, speaking duration, and the number of utterances are culture-dependent. It has been observed that people from Western countries, such as the USA and Germany, tend to avoid silence during a conversation, while people from Eastern countries, such as India and Japan, appreciate it [75].

Finally, this study only used a limited number of audio features to predict well-being, which does not comprehensively capture the full spectrum of communication processes in a teamwork session. Furthermore, individual well-being in a team is influenced not only by team processes such as communication but also by input variables such as team composition and other team processes such as cohesion.

Although all best practices were followed to avoid overfitting, too few data samples were available to train reliable and robust machine learning models. Consequently, the accuracy of the models on the validation set fluctuated depending on the training split used. It should be noted that although the final accuracies of the two-class version had an average accuracy of 62.71% per pillar, which was better than the baseline accuracy of 50%, the performance was still not sufficient to reliably predict well-being.

In addition, SHAP values do not provide causal information about the relationship between features and target variables. Instead, they serve to identify features that strongly influence the predictions of the model. Although this information may suggest causality in some cases, it should be used with caution because it does not necessarily prove it.

## 6. Future Work

To enhance the overall robustness of speaker diarization, multiple state-of-the-art systems could be fused by applying the diarization output voting error reduction (DOVER) method [76], which combines multiple diarization results based on a voting scheme. To obtain a quantitative measure of the built speaker diarization system, it could be evaluated based on the diarization error rate (DER) using standard datasets, such as VoxCeleb [23], VoxCeleb2 [24], or AVA-AVD [26]. These datasets provide a benchmark for speaker diarization systems and would allow for a comparison of the performance of the developed system with state-of-the-art approaches.

Moreover, VER could be improved by adding visual information [77] and by overcoming challenges in the areas of robustness, generalization, and usability [78].

Since well-being cannot be predicted from audio data alone, additional team processes and inputs based on the input-process-output (IPO) model could be included to improve prediction quality. This would require deriving new features that could be extracted from the same dataset, e.g., video features as used by Müller [79]. New features could also be calculated by using state-of-the-art speech recognition methods such as Whisper [80] to extract linguistic features. The inclusion of these additional features could potentially improve the performance of the model and deepen our understanding of the factors that contribute to well-being.

However, having more features also requires a more sophisticated feature selection. The RFE feature selection algorithm might lead to biased information loss in the input domain. Even though we addressed this issue by implementing cross-validation and L2 regularization—see Section Feature Extraction from Time Series Data—we acknowledge that potential biases still might occur, which could be improved with more sophisticated selection techniques.

To reduce variability and increase model accuracy, it is necessary to obtain a larger number of data samples. Therefore, it is necessary to conduct additional experiments that are similar in nature and combine the resulting datasets with those collected in this study.

## 7. Conclusions

In conclusion, this study aimed to predict individual well-being based on audio data from teamwork sessions, focusing on identifying the best method and understanding which speech features are the most important predictors. This research suggested that classification methods, particularly two-class classification, are best for predicting well-being based on the PERMA framework. Different machine learning models, such as XGBoost and k-NN, were shown to be optimal for different PERMA pillars.

The study identified important features for the prediction of each PERMA pillar, with the standard deviation of arousal being the most important factor for well-being. A higher variation in the arousal of a person's voice was related to higher perceived well-being. Further research could investigate the causality between these feature relationships. However, this research showed the potential for using speech features as proxies for automatically predicting well-being without the need for surveys. This is an important step toward virtual mirroring, allowing individuals to improve their happiness.

**Author Contributions:** Conceptualization, P.A.G.; methodology P.A.G. and G.J.H.; software, T.Z.; formal analysis, T.Z. and M.M.; investigation, T.Z. and M.M.; resources, I.V. and P.A.G.; data curation, T.Z. and M.M.; writing—original draft preparation, T.Z.; writing—review and editing, T.Z., G.J.H., P.A.G. and I.V.; visualization, supervision P.A.G., I.V. and G.J.H.; project administration, I.V. and P.A.G.; funding acquisition, I.V. All authors have read and agreed to the published version of the manuscript.

**Funding:** This work was supported by a fellowship of the German Academic Exchange Service (DAAD).

**Institutional Review Board Statement:** This study was approved by MIT COUHES under IRB 1701817083 dated 19 January 2023.

**Informed Consent Statement:** Informed consent was obtained from all subjects analyzed in this study. All students who did not sign an informed consent form were excluded from the feature extraction and analysis by using our visual speaker identification pipeline or manual intervention if needed.

**Data Availability Statement:** Data are unavailable due to privacy restrictions.

**Acknowledgments:** We thank Bryan Moser for generously supporting our experiments, and Luis De la Cal for his assistance in running the experiments.

**Conflicts of Interest:** The authors declare no conflicts of interest.

## Abbreviations

| | |
|---|---|
| AIA | Anonymous Institute |
| AS | Anonymous Study |
| ASD | active speaker detection |
| CatBoost | categorical boosting |
| DBSCAN | density-based spatial clustering of applications with noise |
| fps | frames per second |
| IOU | intersection over union |
| IPO | input-process-output |
| k-NN | k-nearest neighbor |
| LOF | local outlier factor |
| LOOCV | leave-one-out cross-validation |
| MAE | mean absolute error |
| PERMA | positive emotion, engagement, relationships, meaning, and accomplishment |
| RFE | recursive feature elimination |
| RTTM | rich transcription time marked |
| S3FD | single shot scale-invariant face detector |
| SHAP | Shapley additive explanations |
| VER | voice emotion recognition |
| XGBoost | extreme gradient boosting |

**Appendix A**

This section answers the question, how the satisfaction scores should be classified, i.e., if it should be a two-class classification problem (satisfactory vs. non-satisfactory), a three-class classification problem (satisfactory vs. neutral vs. non-satisfactory), or a four class classification problem. To determine the optimal number of classes, the ratio between prediction and baseline scores for the different versions was calculated for each pillar (see Figure A1). Each version performed differently for different pillars.

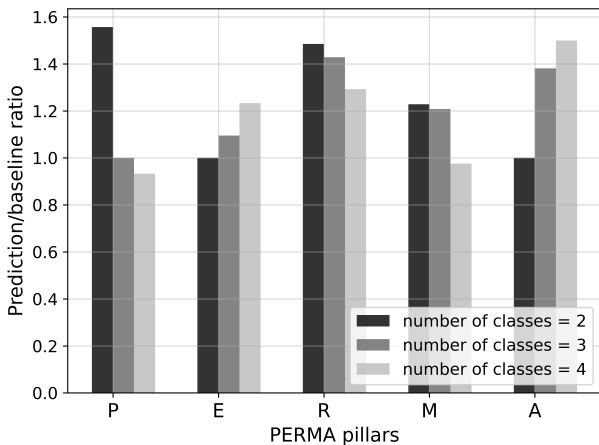

**Figure A1.** Comparison between the prediction baseline ratios for the versions with the different number of classes (higher is better).

For example, the two-class version performed best for pillars P and R, whereas the three-class version performed best for pillars R and A. Almost all ratios were above 1, indicating better performance than the baseline. The average ratio across all pillars was then calculated to determine the best performing version. The results showed that the two-class version performed best with an average ratio of 1.25. In contrast, the three-class version had an average ratio of 1.22, and the four-class version of 1.19.

The comparison was based on the percentage improvement compared to the baseline, which is a metric-independent measure of performance. For classification, the two-class version outperformed the three-class and four-class versions, with an overall improvement of 25.41%. Thus, two-class classification was found to be the best method.

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
