# Peer review of "Predicting Individual Well-Being in Teamwork Contexts Based on Speech Features"

_information, doi:10.3390/info15040217_

Round 1

Reviewer 1 Report (Previous Reviewer 1)

Comments and Suggestions for Authors

Thanks to the authors for their efforts. Most of my concerns were addressed. One minor issue is that, according to the authors, they use cross-validation, but lack a detailed representation of how this is done.

Author Response

Reviewer 2 Report (New Reviewer)

Comments and Suggestions for Authors

The paper proposes a method for predicting individual well-being in teamwork situations using speech features extracted from audio and video data, and using machine learning models to analyze these features.

It would be beneficial to include a specific analysis that directly addresses the added value of video recordings. This could include a comparison of model performance using audio-only features versus features extracted from combined audio and video data.

The criteria for selecting certain speech features should be more clearly justified.

Don't you think that using one 360-degree camera and one microphone per team could introduce biases in data collection, such as uneven audio capture depending on the team member's position in relation to the microphone?

Author Response

Reviewer 3 Report (New Reviewer)

Comments and Suggestions for Authors

Based on the references alone, this field has a rich literature.  However, I did not see citations to the Special Section of the IEEE Signal Processing Magazine on Recent Advances in Affective Computing, Vol. 38, N0. 6, Nov. 2021.  The first 5 articles appear relevant to the current manuscript.  Very specifically, Yu, Li, and Zhao, "Facial-Video-Based Physiological Signal Measurement, pp. 50 ff.  The authors should put this manuscript in the context of those 5 papers and their references.

Comments on the Quality of English Language

None.

Round 2

Reviewer 2 Report (New Reviewer)

Comments and Suggestions for Authors

The article has been corrected appropriately. All the review comments have been taken into account.

Author Response

Reviewer 3 Report (New Reviewer)

Comments and Suggestions for Authors

My prior comments included two papers in the Special Section of the November 2021 issue of the IEEE Signal Processing Magazine that you have ignored:

Alisamir and Ringeval, On the Evolution of Speech Representations for Affective Computing

and

Lee, et al, Deep Representation Learning for Affective Speech Signal Processing and Analysis

These are clearly related to your work and have much good information about speech features which are central to your manuscript.

Comments on the Quality of English Language

NA.

Author Response

This manuscript is a resubmission of an earlier submission. The following is a list of the peer review reports and author responses from that submission.

Round 1

Reviewer 1 Report

Comments and Suggestions for Authors

The manuscript establishes a database of teamwork scenarios for predicting individual well-being. Multiple acoustic features are processed to prepare the dataset and conduct analyses addressing challenges, algorithms, and characterization factors. The signal processing methods are detailed, and the results of evaluating the PERMA pillars are presented using SHPA values and accuracies. Some comments are provided below.

1. It's unclear whether the features listed in Table 2 are inputs to the predictive classifier; this information is not disclosed in the manuscript.

2. Features selected through recursive feature elimination algorithms might be challenging for the classifier to model due to the unique stochastic information in the given feature set. Consequently, applying these features to identify the best classifier for each pillar in Figure 3 may introduce bias.

3. The baseline accuracy is calculated based on two categories of objectives for each pillar. The manuscript lacks clarification on the definitions of these two classes. Additionally, the two-category setup differs from the formulation in Section 3.4.2, which indicates the use of four chance levels in the evaluation. The authors should explain the reason for using two inconsistent baseline systems.

4. In addressing RQ3 in Section 5.1, the experimental results don't convincingly describe or support the notion  from authors that larger fluctuations in intensity levels are a crucial factor in promoting overall well-being.

5. Besides the features used in the manuscript, many other audio features have been applied to emotion recognition tasks. Therefore, the study should provide a detailed explanation of the mechanism used for determining the applied features.

6. Including a brief background on the participants in the manuscript would be beneficial.

Reviewer 2 Report

Comments and Suggestions for Authors

The paper proposes a method to automatically derive social signals related to well-being from raw audio and video data in teamwork contexts. The goal is to develop computational methods and measurements to mirror individuals' well-being to themselves. 

The study challenges current methods for assessing individual well-being in team collaboration that rely on manual surveys.

The study focuses on speech behavior and uses a multi-modal extraction toolchain to predict well-being scores based on speech features extracted from videos using machine learning. To achieve this goal, the study uses a noise-robust multi-modal speaker diarization tool and a curated dataset of teams collaborating on a creative task.

The results suggest that the proposed toolchain can predict individual well-being in teams, leading to better teamwork and happier team members.

I have several minor comments:

The paper does not provide specific information about the background of the participants in the experiment. It mentions that the data was collected from 22 teams with 56 participants collaborating over a four-day period in a team project. However, it does not provide details about the demographics, cultural backgrounds, or any other specific characteristics of the participants.

Further, the number of teams participating in the experiment is not clear. The experiment involved a total of 22 teams (and this is the stated number in the Abstract). However, data was not collected from two teams at the team level, resulting in data from 20 teams. Therefore, the abstract should specify that 20 teams participated in the experiment.

Title and goals: I am not sure the title is accurately mirroring the study when using "Based on Multi-Modal Speech Features". I sugest omiting "multi modality".

Second, the primary objective of this study is to determine computational methods and measurements to facilitate the implementation of virtual mirroring in the context of employee well-being. Employee well-being has gained increasing importance, as acknowledged by the World Health Organization (WHO) classifying burnout as a medical condition resulting from chronic workplace stress. However, the experiment described in the paper was conducted in an academic setting with student participants. It was part of a yearly workshop where students form teams to work on projects for four days, with daily milestones determining team performance.

One can argue for the scenario to be a working place, since the participants were students who were working on team projects in a workshop setting.

I disagree.

In Line 484: “in a real-world working scenario” - I disagree with the example (e.g.). The collaboration is natural for a lab task, and not a real world work scenario. The authors refer to this limitation on Section #5.2, saying that "...the results cannot be generalized due to the limited sample size and the specific group of participants." This is a prominent limitation in the current research, which aims to learn about teamwork behaviour in the same work place.

Lines 83 “...are not scalable for many teams” - The education domain is quite rich with such in-the-wild collecting of material (which is aiming to replace the qualitative teacher-students observations in classes). Could you mentioned at least one prominent research project from this domain?

Lines 169-170: I am not sure I understand the settings. You had an audio recording device (TKGOU) on the table, a 360° camera, in addition laptops of the participants and another laptop that had the Zoom meeting open? And with this you aim “To minimize the intrusiveness“ (line 157)?

Line 193: Which MP4? The one from Zoom or the one from the 360 camera?

Lines 194-195: Please add minimum duration of collaboration and maximum duration of collaboration and the standard deviation of the duration of all teams.

Comments on the Quality of English Language

Line 78: Add a reference to the original project who designed this “usual survival task”.

Line 138: What do you mean “joined”? What were they told about this experiment?

Line 141: "voluntary"  - I believe mentioning here that they had to fill an informed consent will make it more clear.

Line 143: “a monitor” What does that mean? What was the use of this monitor? This term is not mentioned again in the paper.

Line 174: Please refer to the fact that the participants in this experiment are not from the same work place and still you gave them a questionnaire that is “work-related”.

Lines 182-184: Did all of them signed the form of consent?

Lines 183-184: Please correct the English phrasing here. I think you mean that data was not collected from two *participants* in the same team. Right?

Line 267: The authors did not specify whether the speaking duration refers to the entire session or per turn.

Line 268: Relative to what? plese indicate if it refers to calculating the values of the speech features in relation to the specific individual, the team, or both.

Line 271: “b) the interrupter speaks longer than the interrupted” - Do you mean the interrupter “grabbed the floor”. Right? Please use another terminology other than the ambiguous “longer”.

Line 484: I disagree. The collaboration is natural for a lab task, and not a real world work space.

lines 488-489 "Only signals from individuals should be considered who give a privacy consent, ...“ - is not a grammatical sentence.

Page 13, Figure 4, first line: "Class 0" and "class 1" - what do they refer to in the context of the document? These classes are not mentioned before nor after. Are they the two classes represent different levels of well-being? With Class 0 representing a lower level of well-being and Class 1 representing a higher level of well-being?

Lines 547-548: I am not sure how SR can solve the problem of interruptions. Did you mean speaker diarization?

Lines 611-612: Informed Consent Statement : The statement is not what the authors wrote in the paper. Participants who were recorded and video captured did not signed the consent. Only one person from each team signed on it. Therefore “involved” is a matter of definition here. See on lines 350-351: "The students who did not sign an informed consent form were excluded from the feature extraction and analysis by using our visual speaker identification pipeline”.

line 653: References: Why not all items include the year of publication? Is this the journal standard? For example: items 6, 17, and more.